# Nitrogen Removal by Sulfur-Based Carriers in a Membrane Bioreactor (MBR)

**DOI:** 10.3390/membranes8040115

**Published:** 2018-11-22

**Authors:** Thi-Kim-Quyen Vo, Jeong-Jun Lee, Joon-Seok Kang, Seogyeong Park, Han-Seung Kim

**Affiliations:** Department of Environmental Engineering and Energy, Myongji University, 116 Myongji-ro, Cheoin-gu, Yongin-si, Gyeonggi-do 17058, Korea; vokimquyen77@gmail.com (T.-K.-Q.V.); dlwjdwns87@naver.com (J.-J.L.); joonseok0724@gmail.com (J.-S.K.); psk7026@naver.com (S.P.)

**Keywords:** autotrophic denitrification, heterotrophic denitrification, elemental sulfur, membrane bioreactor

## Abstract

Sulfur-based carriers were examined to enhance the nitrogen removal efficiency in a mixed anoxic–anaerobic-membrane bioreactor system, in which sulfur from the carrier acts as an electron donor for the conversion of nitrate to nitrogen gas through the autotrophic denitrification process. A total nitrogen removal efficiency of 63% was observed in the system with carriers, which showed an increase in the removal efficiency of around 20%, compared to the system without carriers. The results also indicated that the carriers had no adverse effect on biological treatment for the organic matter and total phosphorus. The removal efficiencies for chemical oxygen demand (COD) and total phosphorus (TP) were 98% and 37% in both systems, respectively. The generation of sulfate ions was a major disadvantage of using sulfur-based carriers, and resulted in pH drop. The ratio of sulfate in the effluent to nitrate removed in the system ranged from 0.86 to 1.97 mgSO_4_^2−^/mgNO_3_^−^-N, which was lower than the theoretical value and could be regarded as due to the occurrence of simultaneous heterotrophic and autotrophic denitrification.

## 1. Introduction

The increase of nutrient contamination in water sources has been raising a serious problem, due to significant impacts on the aquatic environment, human and animal health. Of the nutrients, nitrogen and phosphorus compounds are the most important, causing eutrophication and serious damage to the ecology of water sources [1,2]. Nitrate in particular is well-known to be able to cause methemoglobinemia, or blue baby syndrome, for infants, and carcinoma, malformation, and mutation when a high concentration is ingested [3,4,5,6]. In the near future, the standards for nutrient discharge in the effluent of wastewater treatment plants will become more stringent in many developed and developing countries. There are various methods for nitrogen removal, including a physico-chemical treatment such as ion-exchange, electrodialysis, precipitation, distillation and reverse osmosis, as well as biological treatment. However, the physical or chemical methods only shift concentrated nitrogen compounds (such as ammonia, nitrite and nitrate) from one liquid phase to another, so a post-treatment is required. On the other hand, the biological method has several advantages over the physical or chemical methods, such as cost-saving, stable and continuous removal of nutrient due to less chemical use, lower energy consumption, and less production of waste solids [7,8].

Nitrogen is removed generally by nitrification and subsequent denitrification in the biological treatment process. First, ammonia is oxidized to nitrite as an intermediate product, and then nitrite is converted to nitrate by autotrophic bacteria (autotrophs) under aerobic conditions. The reduction of nitrate to harmless nitrogen gas takes place under anoxic conditions by heterotrophic bacteria (heterotrophs) or autotrophs [9]. The main advantages of heterotrophic denitrification are high denitrification rate and treatment capacity [10]. When a heterotrophic denitrification supplemented with ethanol was conducted in a membrane bioreactor for drinking water treatment, the removal efficiency for nitrate was up to 92% and nitrite generation was lower (0.35 mg/L) [11]. Since the heterotrophs use organic carbon compounds as an electron donor for their growth and development, the denitrification occurs incompletely when the organic carbon source is insufficient. Most of the wastewaters contained high nitrogen and low carbon concentration (low C/N ratio), such as wastewater from the leather industry, fertilizer factory, landfill leachates and livestock farming, which resulted in the requirement to provide an external carbon source. Because it is very difficult practically to add a correct amount of organic carbon to the denitrification system, a post-treatment is needed to remove residual organic carbon compounds [12,13,14].

In the autotrophic denitrification process, autotrophs such as *Thiobacillus denitrificans* utilize inorganic carbon compounds (e.g., CO_2_, HCO_3_^−^) as their carbon source, and the energy source is derived from oxidation-reduction reactions with hydrogen or inorganic reduced sulfur compounds (H_2_S, S, S_2_O_3_^2−^, S_4_O_6_^2−^, SO_3_^2−^) as the electron donor [15]. In the sulfur-based autotrophic denitrification, elemental sulfur and nitrate react as an electron donor and an acceptor, respectively [16].

55S^0^ + 20CO_2_ + 50NO_3_^−^ + 38H_2_O + 4NH_4_^+^ →4C_5_H_7_O_2_N + 55SO_4_^2−^ + 25N_2_↑ + 64H^+^(1)

Some advantages of autotrophic denitrification include that: (a) there is no external organic carbon source, like ethanol or methanol, which can reduce the cost treatment and poisoning effect of some organic carbon; (b) less biomass is produced, which means sludge production in autotrophic denitrification is two- to threefold less than in heterotrophic denitrification, resulting in the minimization of sludge handling [17,18,19]; (c) elemental sulfur is cheap and readily available, non-toxic, insoluble in water, and stable under normal conditions [20]; and (d) the autotrophic denitrification produces less N_2_O (a greenhouse gas) than heterotrophic denitrification [21]. However, the major disadvantages of this process are sulfate and acid generation, and alkalinity consumption [22]. According to Equation (1), when nitrate is reduced to nitrogen gas, sulfur is oxidized to sulfate and an H^+^ proton is released. As sulfate generation causes the decrease of the pH in the effluent, thus, alkalinity should be supplied in order to maintain a neutral pH in the system. In addition, the specific denitrification rate of autotrophs is lower than that of heterotrophs [18]. Oh et al. reported that a lab-scale sulfur packed column had a complete autotrophic denitrification with the influent concentration of 100 mgNO_3_^−^-N/L and hydraulic retention time of 1.5 h [23].

The combination of sulfur-based denitrification with membrane technology was suggested because autotrophic bacteria have a low specific growth rate, and it is hard to maintain an appropriate amount of biomass within a reactor [24]. In recent years, the membrane bioreactor (MBR) process, which combines biological activated sludge and membrane filtration, has drawn significant attention due to various advantages over conventional activated sludge treatment, such as stable performance, high effluent quality, ease of operation, space saving, high mixed liquor suspended solids (MLSS) concentration, and high sludge retention time (SRT) to maintain biomass concentration at a much higher level. On the other hand, MBR has some drawbacks, such as membrane fouling, and need for membrane cleaning and replacement [25]. A combined process of sulfur-based denitrification and separation by membrane was proposed by Kimura et al., in which the process was operated at a flux of 0.5 m^3^/m^2^/day, hydraulic retention time (HRT) of 160 min, and a biomass concentration of about 1000 mg protein/L, and achieved a complete removal of 25 mgNO_3_^−^-N/L in groundwater [18]. The denitrification performance of a combination of sulfur-utilizing autotrophic denitrification and separation by membrane was evaluated in the study. By installing a membrane, autotrophic denitrifiers, in which growth rates are considerably slow, are kept inside the reactor to increase the nitrogen elimination.

The other studies focused only on the autotrophic denitrification in drinking water to remove nitrate, none of which has been applied to nitrogen treatment in wastewater. In addition, granular sulfur is widely used in the autotrophic denitrification processes. The objectives of this study were to compare the nitrogen removal performance of mixotrophic denitrification to heterotrophic denitrification, and demonstrate the feasibility of a sulfur-based carrier application to increase nitrogen removal effectiveness.

## 2. Materials and Methods

### 2.1. Experimental Set-Up

Experiments were performed in the apparatus illustrated in Figure 1. Two lab-scale systems were operated in parallel with working volume of anoxic, anaerobic and MBR compartments of 2, 3 and 6.9 L, respectively. Two flat sheet membrane modules (width × height = 120 mm × 120 mm) with a surface area of 0.06 m^2^ and pore size of 0.4 µm, from Pure Envitech, South Korea, were submerged in each MBR. Two systems (S1 and S2) were automatically operated by using programmed timers, and the trans-membrane pressure (TMP) was recorded by digital pressure gauges. Synthetic wastewater was pumped directly into the anoxic compartment by using a peristaltic pump to control the feed rate. MBR’s permeate pumps were maintained with intermittent suction (9 mins in operation and 1 min standby). Two membrane bioreactors were operated at flux of 8 L/m^2^/h (LMH). Nitrate in the internal recycle stream from the MBR fed to the anoxic compartment. Aeration was served by air diffusers installed at the bottom of MBRs in the rear end of membrane modules, in order to supply dissolved oxygen for aerobic conditions, and to control the membrane fouling by air scouring. Stirring motors were installed in the anoxic and anaerobic compartments to keep biomass suspended. Dissolved oxygen (DO) concentration was maintained in anoxic, anaerobic, and MBR compartments at values of 0.4, 0.1 and 4 mg/L, respectively. When TMP was higher than 20 kPa, membrane modules were externally cleaned with 0.5% of NaOCl for 6 h prior to use. 

### 2.2. Synthetic Wastewater, Seed Sludge and Sulfur-Based Carriers

The synthetic wastewater was made with 137.5 mg/L C_6_H_12_O_6_, 282.14 mg/L NH_4_HCO_3_, 21.94 mg/L KH_2_PO_4_, 15 mg/L MgSO_4_·7H_2_O, 0.09 mg/L MnSO_4_·H_2_O, 0.3 mg/L ZnSO_4_·7H_2_O, 55 mg/L CaCl_2_·2H_2_O, 3 mg/L FeCl_2_·2H_2_O, 300 mg/L NaHCO_3_. The concentration of synthetic wastewater is chemical oxygen demand (COD) of 141 ± 2.21 mg/L, total nitrogen (TN) of 51 ± 0.90 mg/L, NH_4_^+^-N of 49.24 ± 0.85 mg/L and total phosphorus (TP) of 5.10 ± 0.10 mg/L.

The seed activated sludge was collected from a wastewater treatment plant located in Yongin, South Korea. The sludge retention time was maintained at 40 days during operation periods in both systems. To control SRT, the volume of suspended sludge was withdrawn daily from reactors. MLSS concentration in reactors was maintained at 4526 ± 196 mg/L during operation periods.

The carriers were composed of sulfur (S), calcium carbonate (CaCO_3_) and powder activated carbon (PAC). The elemental sulfur is cheap, readily available, insoluble in water, stable under normal conditions, and easy to handle and store. For these reasons, elemental sulfur was chosen as an electron donor for an autotrophic denitrification in this study. The autotrophic denitrification decreases pH value of the reactor, thus, CaCO_3_ served as pH buffer and PAC was used to improve the biofilm formation on carriers. Sulfur, CaCO_3_, and PAC were mixed with a suitable ratio, and sodium alginate was added to the mixture to bond the components together. The mixture was dried at 105 °C for 24 h, and then broken into pieces. Carriers were added in the anoxic compartment of the first system (S1) at a rate of 1 g/day while the second system (S2) was operated without carriers. In this study, four types of carrier with different percentages of sulfur (C1, C2, C3, C4 with 22%, 27%, 32%, and 47% of sulfur, respectively) were used in order to indicate the nitrogen removal capacity. The percentage of PAC in carriers was kept unchanged at 10%. 

### 2.3. Analytical Methods

Parameters such as COD, TN, NH_4_^+^-N, NO_2_^−^-N, NO_3_^−^-N, TP, and SO_4_^2−^ were determined according to standard methods [26]. TMP values were daily recorded by digital pressure gauges. Ultraviolet absorbance (UVA_254_) value was measured by an Ultraviolet photometer (UV-2800, Shimadzu, Kyoto, Japan).

## 3. Results and Discussion

### 3.1. Removal of Nitrogen

Table 1 summarizes the average concentrations of ammonia, nitrite, nitrate and total nitrogen in the effluents. Combining anoxic and oxic units was studied for nitrogen removal in wastewater treatment. In the MBR zone (an oxic compartment), ammonia (NH_4_^+^-N) was oxidized to nitrite (NO_2_^−^-N), and then converted to nitrate (NO_3_^−^-N), a process known as nitrification. The average concentration of NH_4_^+^-N in influent was 49.24 ± 0.85 mg/L, while it was almost 0 mg/L in two effluents (Table 1). This indicated that ammonia was converted completely, and that the NH_4_^+^-N removal efficiency was 100% in both systems. The specific nitrification rates (SNRs) were approximately 0.018 mgNH_4_^+^-N/mgMLSS/day in both systems, and there was no significant change in SNR value in the first system (S1) with different types of carrier. These results indicated that carriers did not affect the nitrification process.

The denitrification in which nitrate is converted to nitrogen gas takes place in the anoxic compartment. In the second system without carriers (S2), nitrogen in wastewater was removed only by the heterotrophic denitrification process. Heterotrophic denitrifying bacteria derive the required electrons from an organic carbon source that expressed as COD [8], due to the rapid decrease of COD concentrations in anoxic compartments. However, not all nitrate can be removed by the heterotrophic denitrification process if the wastewater has a deficient COD concentration. According to Fu et al., the biological nitrogen removal efficiency was highly affected by the influent COD/N ratio, so that the percentage of nitrogen removal was around 90% at COD/N ratio of 9.3, while it decreased from 71% to 69% when COD/N ratio decreased from 7 to 5.3, respectively [27]. The influent COD/N ratio was only 2.8 in this study, which means that an insufficient organic carbon source was provided for full denitrification to take place, and the NO_3_^−^-N concentration of the permeate was high of 27.3±1.7 mg/L. However, in the first system (S1) with carriers in anoxic compartment, the elemental sulfur of the carrier acted as an electron donor for the conversion of nitrate to nitrogen gas (as Equation (1)). Therefore, not only an autotrophic denitrification but also a heterotrophic denitrification could be considered to occur in S1. The effluent nitrate concentration of S1 was reduced by stimulating the simultaneous heterotrophic and autotrophic denitrification. The specific denitrification rate (SDNR) of S1 was 0.011 mgNO_3_^−^-N/mgMLSS/day, which was higher than that of S2 (0.007 mgNO_3_^−^-N/mgMLSS/day). The SDNRs of S1 with C1, C2, C3, C4 were 0.009, 0.010, 0.010 and 0.011 mgNO_3_^−^-N/mgMLSS/day, respectively.

As a result of conversion from nitrate to nitrogen gas, nitrogen in wastewater is removed. Consequently, the effluent total nitrogen concentrations of both systems were decreased (Figure 2). The first, second, third and fourth periods correspond to carrier C1, C2, C3, C4 added to the anoxic compartment of the first system (S1), while the second system (S2) was without carriers during operation periods. TN removal efficiencies of S1 at various types of carriers from C1 to C4 were 54 ± 4%, 56 ± 2%; 58 ± 2%; 63 ± 3%, in that order. In addition, the specific nitrogen removal rates in the first system were 0.009, 0.010, 0.010 and 0.011 mgTN/mgMLSS.day at C1, C2, C3, C4, respectively. There was a difference in total nitrogen removal efficiency between types of carriers, because of increasing the percentage of sulfur in carriers (from 22% to 47%). When sulfur content increased, more elemental sulfur reacted with nitrate to carry out the denitrification. The TN removal efficiency increased by around 20%, and the specific TN removal rate also increased from 0.008 to 0.011 mgTN/mgMLSS/day, compared to the second system without carriers. It can be said that heterotrophic coupling with autotrophic denitrification by adding carriers enhanced the nitrogen removal in wastewater treatment (Figure 3).

Nguyen et al. indicated that the TN removal efficiency of Sponge-MBR, where heterotrophic denitrification occurred, was 51 ± 18% with hydraulic retention time (HRT) of 10 h, SRT of 20 days, and a specific TN removal rate of 0.011 mgTN/mgMLSS/day [28]. Another study by Kim et al. showed the nitrate treatment performance of sequential heterotrophic and autotrophic denitrification processes, in which two columns connected together were used as reactors for each process to treat 500 mg/L of nitrate in synthetic wastewater. The experimental results indicated that the total nitrate removal efficiency was 96 ± 5%, of which 49 ± 12% was of heterotrophic denitrification and 46 ± 12% was of autotrophic denitrification [29]. Additionally, methanol was added as an organic carbon in a packed-bed bioreactor filled with sulfur, limestone, and activated carbon particles so that autotrophic and heterotrophic denitrification occurred simultaneously and the complete denitrification of 75 mgNO_3_^−^-N/L was achieved [10].

### 3.2. Sulfate Generation

Sulfate (SO_4_^2−^) is one of the end products of a sulfur-based autotrophic denitrification. The variation of sulfate concentrations for different types of carrier is given in Figure 4. The average influent sulfate concentration was 17 ± 2 mg/L. The second system, with only a heterotrophic denitrification, did not produce sulfate, so the effluent sulfate concentration was similar to the influent, while the sulfate concentrations increased in the first system as the percentage of sulfur in carriers increased (Figure 4). The average effluent sulfate concentrations of S1 for carriers C1, C2, C3, C4 were 44 ± 1, 48 ± 2, 55 ± 2 and 75 ± 7 mg/L, respectively. More sulfate production indicated that more nitrate was removed. According to the stoichiometric equation (Equation (1)), 7.54 mg SO_4_^2−^ is produced when 1 mg NO_3_^−^-N is removed. The ratios of the effluent sulfate to the removed nitrate in the first system for various carriers C1, C2, C3, C4, were 0.86, 0.89, 1.35 and 1.97 mgSO_4_^2−^/mgNO_3_^−^-N, respectively. The ratios from the experiments were lower than the theoretical value above-mentioned, which means that heterotrophic and autotrophic denitrification occurred simultaneously [23]. Oh et al. also reported that under mixotrophic conditions, this ratio was 3–4 mgSO_4_^2−^/mgNO_3_^−^-N for denitrification of 100 mg/L nitrate [23]. The lower ratio indicates that less elemental sulfur was utilized to remove unit amount of NO_3_^−^-N, resulting in the reduction of the treatment cost.

### 3.3. COD and TP Removal

The COD concentration of feed and permeates are shown in Figure 5. The average influent COD concentration was maintained at 140 ± 2 mg/L, and the COD concentration in permeates was almost below 5 mg/L even at different operation periods. The average removal efficiency and specific removal rate were 98 ± 2% and 0.049 ± 0.003 mgCOD/mgMLSS/day in both systems during operation periods, respectively. There was not a significant difference in term of COD removal for both systems with different types of carriers. These results indicated that carriers have no effect on biological treatment of organic matters. In comparison to the results of Wen et al., the COD removal efficiency of a conventional MBR achieved was only 80%, and the COD permeate concentration was lower than 30 mg/L [30]. Nguyen et al. also reported that the COD in the permeate of a conventional MBR operated at flux of 6 LMH was always 11–16 mg/L, with 84 ± 10% of COD removal [28]. It was noticeable that most of COD was biodegraded in anoxic tanks. COD acts as an organic carbon source for heterotrophic denitrification processes occurred in anoxic compartments as shown in Figure 6. As a result, the COD concentration of both anoxic compartments decreased rapidly. It was clear from these data that both systems could provide a significant removal of COD in this study.

The same behavior in the system performance could be observed for TP removal. Variation of TP concentration during operation periods is given in Figure 7. The influent TP concentration was maintained at 5 mg/L. The average TP concentrations in permeates of S1, S2 were obtained at 3.2 ± 0.3 and 3.1 ± 0.3 mg/L, respectively. The TP removal efficiency and specific removal rate were around 37 ± 4% and 0.0004 mgTP/mgMLSS/day in both systems, respectively. As in terms of COD removal, the carriers could not improve the total phosphorus removal. The TP removal efficiency of this study was higher than the result of Nguyen et al., who reported that TP removal efficiencies of a conventional MBR and a Sponge-MBR operated at flux of 6 LMH, HRT of 8 h were 20 ± 15% and 26 ± 11%, respectively [28]. This was caused by an anaerobic compartment installed in the system. In the anaerobic tank, poly-phosphorus was hydrolyzed to release ortho-phosphorus, thus the TP concentration in the anaerobic compartment was higher, while ortho-phosphorus is usually accumulated within bacterial cells and released out of system by wasting sludge [8]. Suspended sludge was withdrawn from the membrane bioreactor daily to maintain SRT at 40 days and to eliminate total phosphorus. 

### 3.4. Membrane Fouling

Although the MBR system offers several advantages over the conventional activated sludge process, the membrane fouling is the major disadvantage. Figure 8 shows the changes in TMP of both MBRs during the operation periods. There was no significant difference in TMP development between both systems. The TMP increased from 3.7 to 12 kPa in S1 and from 2.8 to 9.3 kPa in S2 during 39 days (from the second to fourth period). The fouling rates of the two systems were observed to be 0.21 kPa/day for S1 and 0.17 kPa/day for S2. The carrier debris attached on the membrane surface could be thought to be the reason why the higher TMP appeared in S1 than S2 without carriers. Additionally, ultraviolet absorbance was measured for the two systems. The average UVA_254_ values in the effluents of S1 and S2 (0.033 ± 0.008 1/cm and 0.034 ± 0.007 1/cm) were lower than those of the influent (Figure 9). This indicated that the soluble organic matters, especially double bond linkage substances, were trapped in membranes causing the irreversible fouling. In comparison with the later periods, the value of UVA_254_ was lower at the first period. This means that more soluble organic compounds were absorbed on the membrane surface and could cause the higher value of TMP.

## 4. Conclusions

It was clear that the sulfur-based carrier played a significant role in the nitrogen removal in the wastewater treatment. Firstly, the carrier induced the occurrence of a simultaneous autotrophic and heterotrophic denitrification to enhance the nitrogen removal efficiency. Secondly, it is favorable to apply the carrier in wastewater treatment systems to treat the influent with a low C/N ratio, as it has no need of an external carbon source. Thirdly, the actual ratio of sulfate generated to nitrate removed was lower than that of theoretical ratio. The results also indicated that the carriers containing more elemental sulfur achieved a higher nitrogen removal. Based on the results from this study, the evaluation of MBR systems using real wastewater will be required in further studies.

## Figures and Tables

**Figure 1 membranes-08-00115-f001:**
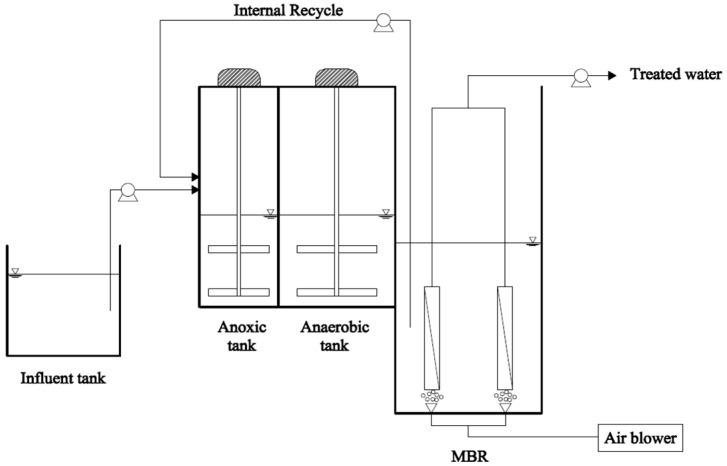
Schematic diagram of a lab-scale system.

**Figure 2 membranes-08-00115-f002:**
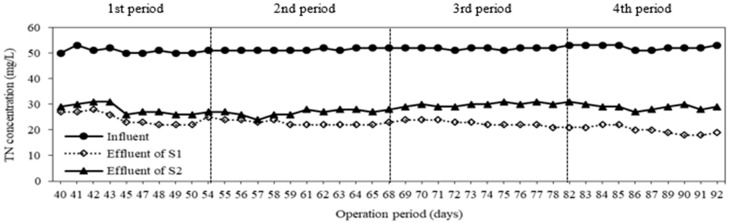
Variation of total nitrogen concentration at different operation periods.

**Figure 3 membranes-08-00115-f003:**
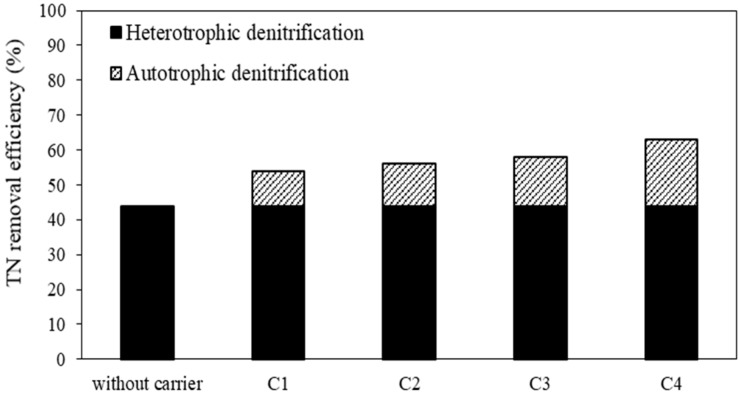
TN (Total Nitrogen) removal efficiency of different types of carrier.

**Figure 4 membranes-08-00115-f004:**
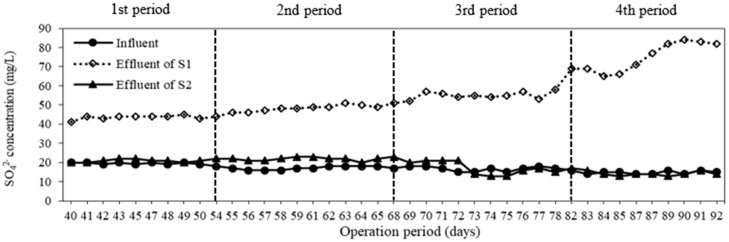
Variation of sulfate concentration at different types of carrier.

**Figure 5 membranes-08-00115-f005:**
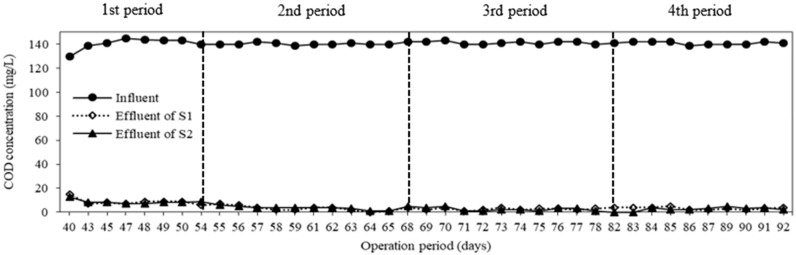
Variation of COD (Chemical Oxygen Demand) concentration at different operation periods.

**Figure 6 membranes-08-00115-f006:**
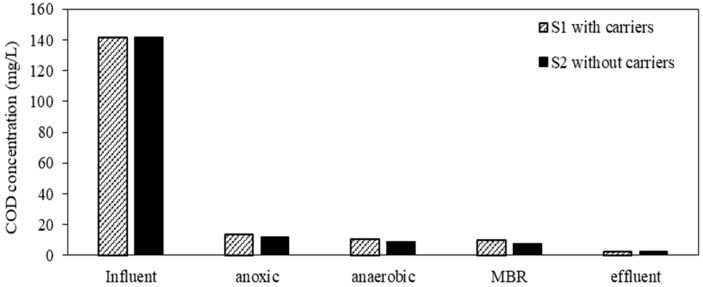
Variation of COD concentration in compartments.

**Figure 7 membranes-08-00115-f007:**
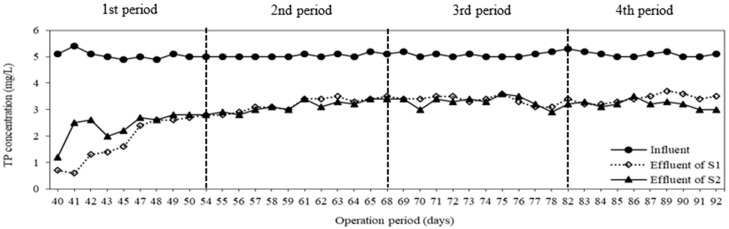
Variation of TP (Total Phosphorus) concentration at different operation periods.

**Figure 8 membranes-08-00115-f008:**
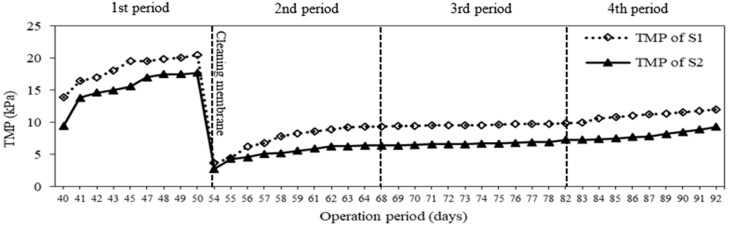
Evolution of TMP (Trans-Membrane Pressure) in two systems at different operation periods.

**Figure 9 membranes-08-00115-f009:**
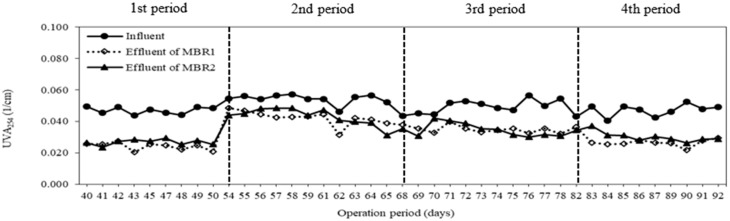
UVA_254_ value during operation period.

**Table 1 membranes-08-00115-t001:** Nitrogen species in the effluents.

Nitrogen Species	S1 with Carriers				S2 without Carriers
	C1	C2	C3	C4	
NH4+-N (mg/L)	0	0	0	0	0
NO2−-N (mg/L)	0.02 ± 0.01	0.04 ± 0.02	0.05 ± 0.01	0.14 ± 0.02	0.06 ± 0.02
NO3−-N (mg/L)	23.5 ± 2.3	22.3 ± 1.1	22.2 ± 1.0	19.5 ± 1.5	27.3 ± 1.7
TN (Total Nitrogen) (mg/L)	24 ± 2	23 ± 1	22 ± 1	19 ± 1	28 ± 2

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
