# Peer review of "Nitrogen Removal by Sulfur-Based Carriers in a Membrane Bioreactor (MBR)"

_membranes, 2018, doi:10.3390/membranes8040115_

Round 1
Reviewer 1 Report
The work is clear and well written, however unclear the real novelty respect to the systems already present. A better comparison with literature in terms of efficiency must be proposed.
The main concerns is about the fouling mechanism. The cleaning methods is not useful to solve the problem and a real solution to limit this phenomenon is not presented.
Probably, describing the mechanism of fouling will help in understanding how to limit it.
After this clarification the manuscript can be published
Author Response
Response to Comments of Reviewer 1
Article: Nitrogen Removal by Sulfur-Based Carriers in Membrane Bioreactor (MBR)
We are very grateful to the reviewer for the valuable suggestions and the constructive comments, which have helped us to improve the quality of the manuscript. Please see our response below.
1. The work is clear and well written, however unclear the real novelty respect to the systems already present. A better comparison with literature in terms of efficiency must be proposed.
→ We are grateful for the reviewer’s comment. We have modified the text to indicate the novelty of this study (page 2, lines 86-91).
2. The main concerns is about the fouling mechanism. The cleaning methods is not useful to solve the problem and a real solution to limit this phenomenon is not presented. Probably, describing the mechanism of fouling will help in understanding how to limit it.
→ We thank the reviewer for this comment. NaOCl is widely used to clean membranes and this was reported in some researches. In this study, after membranes were cleaned by NaOCl for 6h, the TMP value decreased from 20 kPa to 2 kPa (Fig. 8). We think that the cleaning method is useful.

Reviewer 2 Report
The article “Nitrogen Removal by Sulfur-Based Carriers in Membrane Bioreactor (MBR)” indicated the positive values about using the sulfur-containing carriers to enable the autotrophic denitrification, to finally increase the denitrification rate for quicker nitrogen removal. The paper gave sufficient information for various water quality parameters to indicate the effect of adding the carriers for the wastewater treatment. The novelty and paper content are good to be accepted. However, there are still some minor problems needing to be addressed before accepting the paper. Here is attached with the detailed comments to be recommended for the paper.
1. In 2.1 Experimental set-up. The experimental procedures to control DO in anoxic and anaerobic compartments should be indicated.
2. In section 2.1, the time when to use NaOCl to clean the membrane modules should be indicated, and thus the people can catch up the time.
3. In line 125 to 128, the construction procedures or properties to make sulfur carriers should be indicated to facilitate the readers to understand the manufacturing procedures.
4. For section 3.2, there should be description about the sulfate production problem since the conclusion indicated that the sulfate in the effluent might cause some problems, but the problems should be described in the section 3.2.
5. In line 262, for the average UVA254 value comparison, these values should be both measured before and after the membrane filtration, to indicate the positive effect of the membrane for the organic separation.
Author Response
Response to Comments of Reviewer 2
Article: Nitrogen Removal by Sulfur-Based Carriers in Membrane Bioreactor (MBR)
We are very grateful to the reviewer for the valuable suggestions and the constructive comments, which have helped us to improve the quality of the manuscript. Please see our response below.
1. In 2.1 Experimental set-up. The experimental procedures to control DO in anoxic and anaerobic compartments should be indicated.
→ Thanks for noticing. The DO concentrations in anoxic and anaerobic compartments were daily measured and controlled by adjusting the speed of stirring motors.
2. In section 2.1, the time when to use NaOCl to clean the membrane modules should be indicated, and thus the people can catch up the time.
→ The membrane modules were cleaned by NaOCl when TMP value was higher than 20 kPa (a recommended operating pressure). The modified paragraph can be found on page 3 (lines 107-108).
3. In line 125 to 128, the construction procedures or properties to make sulfur carriers should be indicated to facilitate the readers to understand the manufacturing procedures.
→ Thanks for noticing. The procedure of making sulfur-based carriers was added on page 3 (lines 125-127).
4. For section 3.2, there should be description about the sulfate production problem since the conclusion indicated that the sulfate in the effluent might cause some problems, but the problems should be described in the section 3.2.
→ We thank the reviewer for this comment. The modified paragraph can be found on page 2 (lines 66-67).
5. In line 262, for the average UVA254 value comparison, these values should be both measured before and after the membrane filtration, to indicate the positive effect of the membrane for the organic separation.
→ We are grateful for the reviewer’s comment. This is a shortcoming of this study. We will measure UVA254 before and after the membrane filtration in further studies.

Reviewer 3 Report
Reviewer’s comments on Manuscript
Title: Nitrogen Removal by Sulfur-Based Carriers in Membrane Bioreactor (MBR)
The reviewer is glad to be invited for reviewing this manuscript. The article is about autotrophic and heterotrophic denitrification by using sulfur-based carriers in membrane bioreactors. The topic is of some interest to the potential readers of the journal. The workload is sufficient for publication. However, the manuscript is generally poorly synthesized, particularly the language. The authors should address quite some issues before accepting this submission for publication. My suggestion is major revision.
Major:
1. First, the authors ought to clarify the significance of the study to the field. What is the novelty of the study?
2. The introduction and conclusion sections should be extensively strengthened. The introduction is not coherent enough, thus reorganization is required. The conclusion is too weak. In addition to synopsize the findings of the study, the authors should also emphasize the importance of the study and make some suggestions for future studies. What is the take-home message?
3. The language of this articles is one of the major issues. The authors should really have the writing retouched by someone capable of competent English Technical Writing.
Minor:
1. The authors do not have to define “denitrification” and “nitrification”. When someone is interested in the topic of the article, they would have known what denitrification is.
2. Why using the given synthetic water recipe? Please cite the literature source or explain its reality relevance.
3. What was the membrane material?
4. What was the pH in the system? Have the authors considered the potential loss of ammonium-nitrogen through ammonia gas under higher pHs?
5. In Figure 3, how did the authors determine the percentage contributions of heterotrophic denitrification and autotrophic denitrification? By simply subtracting the heterotrophic denitrification observed in the system without carriers? Is that technically convincing to assume the heterotrophic denitrification efficiency stays unchanged when the environmental condition varies across different reactors?
6. Have the authors performed statistical calculations to compare the different performance of the systems?
7. The references are generally old. Among the 30 cited papers, only three were published 2014. Nothing later than 2016 is cited in the paper. And it is at the end of 2018.
Author Response
Response to Comments of Reviewer 3
Article: Nitrogen Removal by Sulfur-Based Carriers in Membrane Bioreactor (MBR)
We are very grateful to the reviewer for the valuable suggestions and the constructive comments, which have helped us to improve the quality of the manuscript. Please see our response below.
Major:
1. First, the authors ought to clarify the significance of the study to the field. What is the novelty of the study?
→ We thank the reviewer for this comment. The modified paragraph can be found on page 2 (lines 86-91).
2. The introduction and conclusion sections should be extensively strengthened. The introduction is not coherent enough, thus reorganization is required. The conclusion is too weak. In addition to synopsize the findings of the study, the authors should also emphasize the importance of the study and make some suggestions for future studies. What is the take-home message?
→ We thank the reviewer for this comment. We have modified the introduction and conclusion. The conclusion was revised as below:
“The sulfur-based carrier is an effective solution for the removal of nitrogen in wastewater. Firstly, sulfur-based carrier causes the occurrence of a simultaneous autotrophic and heterotrophic denitrification to enhance the nitrogen removal efficiency. Due to no need an external carbon source, it is favorable to apply in wastewater treatment with a low C/N ratio. Thirdly, the actual ratio of sulfate generated to nitrate removed was lower than that theoretical ratio. The results also indicated that carriers containing more elemental sulfur achieve a higher nitrogen removal. One of the limits in this research is the using of synthetic wastewater, thus, further studies with real wastewater are proposed.”
3. The language of this articles is one of the major issues. The authors should really have the writing retouched by someone capable of competent English Technical Writing.
→ We thank the reviewer for this comment. We tried to correct grammar errors for the whole manuscript.
Minor:
1. The authors do not have to define “denitrification” and “nitrification”. When someone is interested in the topic of the article, they would have known what denitrification is.
→ We thank the reviewer for this comment. The modified paragraph can be found on page 1 (lines 38-42).
2. Why using the given synthetic water recipe? Please cite the literature source or explain its reality relevance.
→ We are grateful for the reviewer’s comment. We used the synthetic water because we wanted to fix the wastewater component. It is easy to compare the nitrogen removal efficiency of systems with or without carriers.
3. What was the membrane material?
→ Thanks for noticing. The membrane material was polyvinyl chloride from the Pure Envitech company in South of Korea.
4. What was the pH in the system? Have the authors considered the potential loss of ammonium-nitrogen through ammonia gas under higher pHs?
→ We acknowledge the reviewer’s suggestion. However, the main objective of this study was to demonstrate the feasibility of a sulfur-based carrier application to increase nitrogen removal efficiency. So, we will focus this mechanism in further studies.
5. In Figure 3, how did the authors determine the percentage contributions of heterotrophic denitrification and autotrophic denitrification? By simply subtracting the heterotrophic denitrification observed in the system without carriers? Is that technically convincing to assume the heterotrophic denitrification efficiency stays unchanged when the environmental condition varies across different reactors?
→ We totally agree with the reviewer’s comment. This is a shortcoming of this study.
6. Have the authors performed statistical calculations to compare the different performance of the systems?
→ Yes, we have.
7. The references are generally old. Among the 30 cited papers, only three were published 2014. Nothing later than 2016 is cited in the paper. And it is at the end of 2018.
→ We are grateful for the reviewer’s comment. However, there are a few references on the treatment of nitrogen in wastewater by using a autotrophic denitrification, especially by sulfur-based carriers. We only updated like as in the manuscript.

Round 2
Reviewer 1 Report
The work was improved and ready for publication
Author Response
Thank you so much.
Reviewer 2 Report
The questions are addressed well, and the paper is good to be accepted.Author Response
Thank you so much.
Reviewer 3 Report
Grammar issues still exist, even in the newly added/revised sentences.
The conclusion is not in a strong shape.
Author Response
We acknowledge the reviewer's comments.
The conclusion and grammar issues were modified.